# A High-Level Representation of the Navigation Behavior of Website Visitors

Alicia Huidobro [1], Raúl Monroy [1,*] and Bárbara Cervantes [2,†]

1   School of Engineering and Science, Tecnologico de Monterrey, Atizapan de Zaragoza 52926, Mexico; a01749803@itesm.mx
2   Virtus Be Real S.A. de C.V., Atizapan de Zaragoza 52987, Mexico; barb@virtus.mx
*   Correspondence: raulm@tec.mx
†   This research was carried while Dr. Cervantes was a Research Assistant at Tecnologico de Monterrey.

**Abstract:** Knowing how visitors navigate a website can lead to different applications. For example, providing a personalized navigation experience or identifying website failures. In this paper, we present a method for representing the navigation behavior of an entire class of website visitors in a moderately small graph, aiming to ease the task of web analysis, especially in marketing areas. Current solutions are mainly oriented to a detailed page-by-page analysis. Thus, obtaining a high-level abstraction of an entire class of visitors may involve the analysis of large amounts of data and become an overwhelming task. Our approach extracts the navigation behavior that is common among a certain class of visitors to create a graph that summarizes class navigation behavior and enables a contrast of classes. The method works by representing website sessions as the sequence of visited pages. Sub-sequences of visited pages of common occurrence are identified as "rules". Then, we replace those rules with a symbol that is given a representative name and use it to obtain a shrinked representation of a session. Finally, this shrinked representation is used to create a graph of the navigation behavior of a visitor class (group of visitors relevant to the desired analysis). Our results show that a few rules are enough to capture a visitor class. Since each class is associated with a conversion, a marketing expert can easily find out what makes classes different.

**Keywords:** web analytics; web log mining; clickstream analysis; sequence mining; sequitur; graph techniques

## 1. Introduction

The more knowledge a company has about visitors, the more effective its marketing strategies will be [1–3]. Therefore, it is valuable to know how visitors navigate the website [4–6]. This knowledge has to be obtained from the huge amount of data that are stored on a website [6–8]. Web analytics solutions (WAS) are widely used and provide useful metrics [9–17]. However, they have some limitations for describing the navigation behavior of visitors. Current web analytics software provides a page-by-page report [11,12]. This level of detail produces huge graphs that are difficult to analyze and compare. Numerous literature approaches analyze the sequence of visited pages [18–22] but they do not provide a high-level description of the navigation behavior. They are limited to cluster visitors based on different criteria; for example, the longest common subsequence of visited pages [20,23–27].

The objective of our research is to find out the navigation behavior that is common in a whole class of visitors. Each class of visitors should provide valuable knowledge in terms of business goals, specifically for marketing experts. Therefore, the segmentation of visitors is important. We used conversions as classes of visitors, as proposed by A. Huidobro et al. [28]. This approach eases the interpretation of results because conversions are specific visitor actions that contribute to business objectives [3,5,29], and it is a concept with which marketing experts

are familiar. Examples of website conversions are: to pay for a product or a service, to fill in a form with contact details, or to post a positive product review.

To describe the navigation behavior of a whole class of visitors, we started by representing sessions as a sequence of visited pages. From the sequences representing sessions of a given class, we extracted the most frequent subsequences of pages. We called those sequences "rules". Each rule is formed by different pages and represents visitors actions. For example, making a payment or searching for the availability of a product. We named the rules and used them to obtain a reduced representation of each session; session reduction is a result of replacing sub-sequences (of length greater than or equal to two) for the name of the corresponding rule. The representation of sessions with rules drastically reduces the length of sessions (for example, from forty pages to two rules). Various types of analyses can be performed with the reduced representation of sequences. For example, comparing the frequency of a given rule in two different classes of visitors. We show the result of our analysis in a graph to facilitate the explanation oriented to marketing experts. Using the described method, we reduce hundreds of nodes and edges into a simplified graph that captures the navigation behavior of a whole class of visitors. The graph also assists in the comparison of the navigation behavior of different classes of visitors. It prevents the marketing expert from analyzing huge graphs to understand the navigation behavior of visitors. Our four-step methodology is shown in Figure 1.

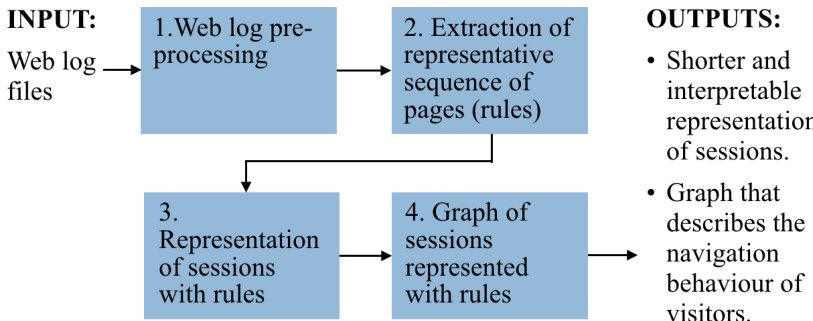

**Figure 1.** The four-step methodology for characterizing website visitors based on their navigation behavior.

## 1.1. Related Work

In this subsection, we explain the limitations of both popular commercial software and literature approaches for describing the navigation behavior of a whole class of visitors. Concerning commercial software, we focus on Google Analytics and Matomo, which have a similar functionality. Google Analytics is the most popular web analytics software [30–32] and Matomo is an alternative to overcome some limitations of Google Analytics [12].

### 1.1.1. Commercial Software

Google Analytics and Matomo provide a similar functionality for tracking the navigation behavior of visitors. In Google Analytics, it is called a "Behavior flow" report. In Matomo, it is the section "Goal conversion tracking", but it is only available in the premium version. Both consist of showing the sequence of the most visited pages in a period. It aims to measure the engagement page to page. Therefore, it is useful for finding pages where the traffic is lost, but it is difficult to follow a path with numerous pages. It is also difficult to visualize the path of 100% of visitors if they are numerous and behave differently. It is possible to track events instead of pages. Nevertheless, those events have to be previously configured. Therefore, events do not represent the natural navigation behavior of visitors. In Figure 2, we show an example of the behavior flow chart in Google Analytics. The page-by-page detail does not provide a high-level description of the navigation behavior [11,12]. Tens of pages would have to be reviewed to understand the navigation behavior of a whole class of visitors.

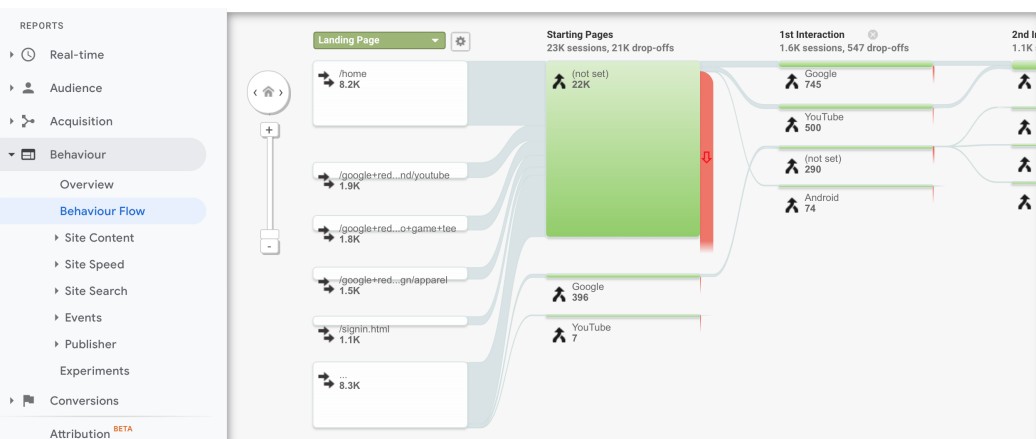

**Figure 2.** Example of the "Behavior flow" report in Google Analytics. It shows the sequence of most visited pages, from left to right. The thick red lines indicate traffic drop-offs. Each column corresponds to a web page or event. Therefore, this type of visualization does not show all the navigation in a single view but continues to the right, according to the number of pages or events.

1.1.2. Other Non-Commercial Approaches

There are diverse web log mining approaches in the literature. However, they are centered around identifying clusters of visitors, and not on obtaining a high-level description of their navigation behavior. The analysis of visited pages, called sequence mining, is commonly applied for discovering patterns with a frequency support measure [33]. Clickstream analysis is the most popular sequence mining approach used for clustering visitors [21]. Clickstream is the sequence of pages visited by a user in a given website and period [27]. In our approach, a rule is a sequence of pages frequently visited by visitors of the same class. Therefore, we reviewed clickstream approaches; below, we describe some of them.

S. Tiwari et al. [20] use previously visited pages to forecast online navigational patterns (finding the next page expectation). They apply agglomerative clustering to group visitors according to the previous web data accessed. They obtain the set of frequently visited pages in each group of visitors. This information is used to put in the cache pages with higher frequency in order to reduce the search time.

A. Banerjee et al. [27] propose finding the longest common sub-sequence of clickstreams using a dynamic programming algorithm. Then, they identify similar users by computing a similarity value that considers the time spent on each page. With the similarity values, they construct a weighted similarity graph. Finally, they find clusters on that graph. They found that, in some cases, there are no exact matches. As a solution, they propose to first group data into categories.

There are other clickstream pattern mining approaches, but they are focused on improving the runtime or memory consumption for clustering visitors [21,22]. Visualization tools have also been proposed to analyze the navigation behavior of visitors [34–39]. However, they provide a detailed analysis of web pages; for example, to find the percentage of visitors on each web page.

Our contribution is a high-level description of the navigation behavior of visitors. A distinguishing characteristic of our approach is that we extracted the natural navigation behavior of visitors instead of finding if visitors perform previously known actions. Another distinctive aspect is that we represented business functions (conversions) in a single node (rule); this data reduction is relevant because representing all of the sessions of an e-commerce website usually involves thousands of visitors and hundreds of pages. In Figure 3, we show an example of sessions represented with rules. Considering that each rule groups N web pages, the proposed representation reduces the information a business expert needs to analyze while keeping interpretability. The representation obtained by commercial software would involve much more nodes (see Figure 2). This would be

difficult, for example, in the identification of loops that are worth analyzing and comparing different entry points to the website. We describe this in more detail in Sections 4 and 5.

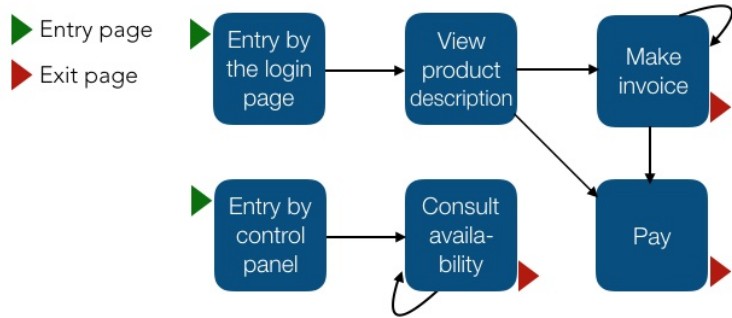

**Figure 3.** An example of sessions represented with rules. Each node represents a rule (business functions) that groups N web pages. We can see, for example, that visitors who arrive at the website by the login page have a greater chance of paying than those who enter through the control panel. We also identify loops. The loop in the "Consult availability" rule may be expected because visitors usually review the availability of N products. However, the loop in the "Make invoice" is worth investigating since it could be a cause of dropout.

*1.2. Methodology*

Web log mining is the use of data mining techniques to obtain information about the navigation behavior of visitors [40]. The main difficulties in web log mining are the huge amount of traffic on websites and the wide variety of paths that visitors could follow [41]. Understanding the navigation behavior of numerous visitors can be overwhelming. Therefore, we aim to describe the navigation behavior of visitors with a simplified representation. With a sequence mining approach, we captured the milestones of different classes of visitors and presented them as a graph. That reduced the amount of data that a marketing expert would have to analyze in order to understand the navigation behavior of visitors and contrast different classes of visitors. To achieve this goal, we represented the navigation behavior of the website visitors as the sequence of visited pages in each session. That representation allows us to find out the representative navigation milestones in each class of visitors. To this aim, we used a compression algorithm that allowed us to identify sequences of pages that are common among visitors from the same class. We called those common sequences "rules". The set of rules obtained in each class of visitors describes the navigation behavior of most of the visitors in that class. Having identified the rules for each class of visitors, it is possible to replace the session pages with rules. This results in a reduced representation of sessions that allows us to carry out different kinds of analyses. For example, sessions could be represented exclusively with rules or, conversely, the behavior that is not represented by rules could be analyzed further. Statistics of the sessions represented as rules would help a marketing expert to establish questions of interest. We analyzed those statistics and, due to our simplification purpose, we found it relevant to represent the navigation behavior of visitors only with the most frequent rules. For our target audience, who are marketing experts, a graph visualization of results would be more friendly. Therefore, we summarized all of the sessions of a given class in a graph. The representation of sessions with rules significantly reduced the amount of data that a marketing expert would have to analyze. The graph depiction assists in the understanding of the navigation behavior of visitors, even when our work was not focused on visualization techniques.

In Section 2, we explain the sequence mining process for identifying rules in each class of visitors. Then, in Section 3, we describe how we used rules for representing sessions. In Section 4, we present the graph that describes the navigation behavior of visitors and insights obtained from it. In Section 5, we summarize our contributions and compare them

with previous work. Finally, in Section 6, we mention the advantages and limitations of the proposed methodology.

## 2. Identification of Rules in Each Class of Visitors

To find out the most common sub-sequences of visited pages for different classes of visitors, in the first place, one needs to represent the navigation behavior of website visitors. With that representation, it is then possible to find sequences of web pages that are commonly visited among visitors from a given class. Below we describe: (1) how we represented sessions with a sequence of symbols that contains the navigation behavior of interest, (2) the compression algorithm that we used to find common sub-sequences of pages, and (3) how we used that compression algorithm to find out the most common sub-sequences of visited pages, which are the milestones for each class of visitors.

### 2.1. Representation of Each Class of Visitors as a Sequence of Symbols

The input data consist of 50,820 sessions represented as the list of visited pages. There were thousands of different pages, but only a small subset was relevant for the proposed analysis. Once we identified relevant pages on sessions, we represented them as symbols to ease the sequence mining process.

#### 2.1.1. Identification of Relevant Pages

We are interested in the navigation behavior that could be meaningful for marketing experts. Therefore, we identified relevant pages, which are described as follows:

1. Filtering of pages of interest: Our marketing partner prepared a list of 298 web pages that they were interested in analyzing. We removed from sessions all web pages that were not pages of interest. We eliminated the sessions that did not visit any page of interest (26%). The dataset was reduced to 37,400 sessions (74%).
2. Removing of pages that are automatically loaded: Those pages are not meaningful for marketing experts because they do not represent the intentional navigation behavior of visitors (for example, Java resources necessary for the proper functioning of the site).
3. Removing the subsequent repetition of the same page: Most sessions had pages subsequently repeated n times. For example, Home → Home → Home → Home → Login → Control panel → Control panel → Logout. Our information technology partner indicated to us that, in most cases, it is due to the functionality of the website, and not related to the navigation behavior of visitors. For example, if the visitor fulfils a form, the same page could be automatically reloaded whenever the visitor clicks on a different field. Therefore, we reduced the subsequent occurrences of the same page to one occurrence. The previous example would be reduced to Home → Login→ Control panel → Logout.

The process to keep only relevant pages entailed some information loss. That information could be useful for some traffic analytics; for example, to measure the number of pages sent to the visitor, the amount of data transmitted, or the frequency of clicks. Nevertheless, that loss does not affect the objective of describing the navigation behavior of visitors. We only needed web pages intentionally visited.

#### 2.1.2. Representation of Sessions as a Sequence of Symbols

To ease the sequence mining process, we represented each session as a sequence of symbols. Due to the fact that there are 298 pages of interest, we assigned a two-letter identifier to each of them. Then, in each session, we replaced the name of the page with its identifier; for example, the session Home → Login→ Control panel → Logout became *AaAzBkBb*.

### 2.1.3. Segmentation of Data in Different Classes of Visitors

We are interested in describing the navigation behavior of different classes of visitors and contrasting them. Therefore, it is necessary to segment data. The input dataset was already labelled. We classified 100% of sessions into four disjoint classes:

- Visitors who made a payment: 7% of the sessions.
- Visitors who started the payment process but did not conclude it: 10% of the sessions.
- Visitors who made a conversion different to the made payment or started payment: 32% of the sessions.
- Visitors who did not perform any conversion: 52% of the sessions.

We will refer to the previous classes of visitors as "Made payment", "Started payment", "Other conversions", and "No conversion". We obtained a dataset for each class of visitors.

With sessions represented as a sequence of symbols and segmented into different classes of visitors, it is possible to identify the representative navigation milestones in each class of visitors. To this aim, we used a compression algorithm, which is described next.

### 2.2. Selection and Implementation of the Compression Algorithm

An objective of our research was to reduce the amount of data that have to be analyzed in order to understanding the navigation behavior of visitors. Our strategy was to find recurrent sub-sequences of visited pages. Therefore, we used a sequence mining approach. In this subsection, we explain how we selected the sequence mining algorithm, how it operates, and the implementation that we used.

### 2.2.1. Selection of the Sequence Mining Algorithm

We discarded algorithms that find the longest common sub-sequence, such as MAXLEN [27,42], because we are interested in all of the sub-sequences that are repeated, no matter if they are long. We evaluated compression algorithms such as Sequitur [42–44], Repair [45,46], and Bisection [42,47]. We selected the Sequitur algorithm because it is the most efficient. It runs in linear time. Below, we explain this algorithm.

### 2.2.2. Sequitur algorithm

Sequitur finds repetitive sub-sequences in a sequence by identifying rules. It creates a grammar based on repeated sub-sequences. Then, each repeated sub-sequence becomes a rule in the grammar. To produce a concise representation of the sequence, two properties must be met [48,49]:

- $p1$ (digram uniqueness) : there is no pair of adjacent symbols repeated in the grammar.
- $p2$ (rule utility): every rule appears more than once.

To clarify the operation of Sequitur, we will use the following definitions:

- Sequence: a string of symbols, e.g., "aghhhhbfababdchdttttyhhs".
- Rule: a sub-sequence that appears twice or more in a sequence and its minimum length is 2. The rules obtained with the Sequitur algorithm may be defined in terms of other rules.
- Base rule: a rule that does not contain other rules, e.g., rule 1 = "a b", rule 2 = "d c".
- Nested rule: a rule composed of base rule(s), e.g., if rule 3 = "f 1 1 2 h", it is a nested rule defined in terms of the base rules 1 and 2.
- Expanded rule: the result of recursively unfolding all the rules that are contained in a nested rule, e.g., the nested rule "f 1 1 2 h" is expanded as "f a b a b d c h".

We will use the sequence aghdfghmadfgh as an example to describe the operation of the Sequitur algorithm. For each symbol in the sequence, Sequitur verifies the properties of digram uniqueness and rule utility. In Table 1, in each row, we show the resulting grammar and the expanded rules, as each new symbol is reviewed. In the column "Resulting grammar", 1 to n are the found rules, and 0 is the result of using those rules in the original string. Grammar 0 is not expanded in the last column because it is not a rule. However, if we expand Grammar 0, we obtain the original string. We can see that Sequitur does not

find any rule from rows 1 to 7. That is because there are no pairs of symbols that appear twice or more in the string. In row 8, the pair of symbols "gh" appears twice, so it is added to the grammar as rule 1. In row 11, the pair of symbols "aa" appears twice and it is added to the grammar as rule. In row 13, the pair of symbols "df" appears twice and it is added to the grammar as rule 3. In row 15, the rule "df" becomes a nested rule because "dfgh" is found twice, but the pair "gh" is already rule 2, so rule 3 changes from "df" to "df 2". All of the rules added to the grammar met properties $p1$ and $p2$.

### 2.2.3. Implementation of the Sequitur Algorithm

We used a publicly available implementation of Sequitur [50]. We adapted this implementation in order to use it with the two-letter identifier of each web page. That was necessary because the original implementation identifies each symbol as a different element in the sequence.

**Table 1.** Operation of the Sequitur algorithm. NRF = No rules found.

| No. | New Symbol | The String so Far | Resulting Grammar | Expanded Rules |
|---|---|---|---|---|
| 1 | a | a | $0 \to a$ | NRF |
| 2 | g | ag | $0 \to a\ g$ | NRF |
| 3 | h | agh | $0 \to a\ g\ h$ | NRF |
| 4 | d | aghd | $0 \to a\ g\ h\ d$ | NRF |
| 5 | f | aghdf | $0 \to a\ g\ h\ d\ f$ | NRF |
| 6 | g | aghdfg | $0 \to a\ g\ h\ d\ f\ g$ | NRF |
| 7 | h | aghdfgh | $0 \to a\ 1\ d\ f\ 1\ 1 \to g\ h$ | gh |
| 8 | m | aghdfghm | $0 \to a\ 1\ d\ f\ 1\ m\ 1 \to g\ h$ | gh |
| 9 | a | aghdfghma | $0 \to a\ 1\ d\ f\ 1\ m\ a\ 1 \to g\ h$ | gh |
| 10 | d | aghdfghmad | $0 \to a\ 1\ d\ f\ 1\ m\ a\ d\ 1 \to g\ h$ | gh |
| 11 | f | aghdfghmadf | $0 \to a\ 1\ 2\ 1\ m\ a\ 2\ 1 \to g\ h$ | gh |
| | | | $2 \to d\ f$ | df |
| 12 | g | aghdfghmadfg | $0 \to a\ 1\ 2\ 1\ m\ a\ 2\ g\ 1 \to g\ h$ | gh |
| | | | $2 \to d\ f$ | df |
| 13 | h | aghdfghmadfgh | $0 \to a\ 1\ 2\ m\ a\ 2\ 1 \to g\ h\ gh$ | |
| | | | $2 \to d\ f\ 1$ | dfgh |

Next, we explain how we used this implementation of the Sequitur algorithm for finding recurrent sub-sequences of visited pages.

### 2.3. Rule Extraction

We used the Sequitur algorithm to find recurrent sub-sequences of visited pages (rules) in each class of visitors. In this subsection, we explain how we extracted, analyzed, and selected those rules.

### 2.3.1. Rule Finding

Sequitur identifies the sub-sequences that appear twice or more in a string as rules. Nevertheless, for our analysis, it was necessary to find all sub-sequences that are common among different sessions. Some of those sub-sequences may appear only once in each session. To this aim, we concatenated sessions of each class of visitors. Below, we explain this methodology.

1.  Concatenate all sessions of a given class of visitors, adding a distinguishing pair of symbols between each session.
2.  Apply the Sequitur algorithm.
3.  Expand rules.
4.  Exclude rules that include the pair of symbols mentioned in the first step.
5.  Compute the frequency of each rule in sessions of the same class of visitors.

In Table 2, we show the percentage of sessions and the number of rules found in each class of visitors. We can see that the classes of visitors "Made payment", "Started payment", and "Other conversions" have a much higher number of rules than the "No conversion" class of visitors, even when "No conversion" has the highest percentage of visitors. This could indicate a more homogeneous behavior in visitors from the first three classes.

**Table 2.** Rules obtained in each class of visitors. Columns 2 to 5 indicate the class of visitor. Seven percent of the sessions are visitors from the class Made payment, where we found 764 rules. Conversely, fifty-two percent of the sessions are visitors from the class No conversion, where we found only 92 rules.

| Metric | Made Payment | Started Payment | Other Conversions | No Conversion |
|---|---|---|---|---|
| Percentage of visitors | 7% | 10% | 32% | 52% |
| Number of rules | 764 | 704 | 997 | 92 |

Rules should allow us to contrast classes. Therefore, we made an inter-class analysis to find out if the set of rules is different in each class of visitors.

### 2.3.2. Inter-Class Analysis

The objective of the inter-class analysis is to find out (1) if the rules are different for each class of visitors, and (2) if those rules are relevant. To this aim, we computed two metrics:

- Percentage of rules found in sessions: given a set of rules, it measures the percentage of those rules that are found in a group of sessions. It allows us to find out if a set of rules describes a specific class of visitors or not. A result of 100% in all classes of visitors for a given set of rules would mean that all those rules were found in the four classes of visitors. Thus, that set of rules would not describe a specific class of visitors.
- Inverse frequency of a rule: it measures the percentage of sessions in which a rule is found at least once. A high percentage indicates that the rule is relevant for describing the navigation behavior of visitors.

As an example of the inter-class analysis, in Table 3, we show the metrics of the rules found in visitors from the class "Made payment". Below, we summarize the interpretation of this table.

- A total of 100% of the rules were found in "Made payment" sessions because rules were extracted from those sessions. We can see that this percentage decreases to approximately 50% for the classes of visitors "Started payment" and "Other conversions". For the visitors from the class "No conversion", it reduces to 6%. These results indicate that approximately 50% of the rules specifically describe the navigation behavior of the visitors that belong to the class "Made payment".
- The highest inverse frequency indicates that one rule was found in up to 91% of sessions that belong to the class of visitors "Made payment". This metric is lower for the other three classes of visitors. Since we use this metric to measure the rule relevance, we can say that this set of rules is more relevant for the visitors that belong to the class "Made payment".

**Table 3.** Rules selected for each class of visitors: the nested rules with inverse frequency ≥5%.

| Metric | Made Payment | Started Payment | Other Conversions | No Conversion |
|---|---|---|---|---|
| Percentage of "made payment" rules found in sessions | 100% | 57% | 46% | 6% |
| Highest inverse frequency of a "made payment" rule in a session | 91% | 68% | 33% | 2% |

We made the inter-class analysis for the other three classes of visitors. Both metrics are higher when the set of rules and the sessions belong to the same class of visitors. Nevertheless, it was remarkable that the highest inverse frequency of No conversion rules was only 14% in the sessions of the same class. This indicates that the behavior of the visitors that belong to the class No conversion is less homogeneous.

The inter-class analysis confirmed that there are relevant and specific rules for each class of visitors. The next step was to select the best rules for describing the navigation behavior of visitors.

### 2.3.3. Rule Selection

A selection of rules is necessary because the rules obtained until now include base rules and nested rules. This is redundant because base rules are contained in nested rules. There could also be rules with too low an inverse frequency (e.g., rules that are found in just one session). These rules are not representative. Therefore, we applied two criteria for selecting rules:

1. Select only nested rules. This eliminates redundancy.
2. Select rules with inverse frequency ≥5%. A rule that describes less than 5% of the sessions does not generalize the navigation behavior of visitors; thus, it is not useful for the objectives of our research.

In Table 4, we show the number of rules obtained after applying these selection criteria. The inverse frequency of all of the nested rules extracted from the class of visitors No conversion was <5%. We concluded that the navigation behavior of this class of visitors is non-homogeneous. Thus, it could not be simplified using a small set of rules. In the next steps of the process, we only used the classes of visitors Made Payment, Started payment, and Other conversions. From now on, when we use the term "rule(s)", we refer to the set of rules presented in Table 4.

**Table 4.** Rules selected for each class of visitors: the nested rules with inverse frequency ≥5%.

| Metric | Made Payment | Started Payment | Other Conversions | No Conversion |
|---|---|---|---|---|
| Number of rules | 9 | 5 | 4 | 0 |

We assigned a name to each rule. In Table 5, we list that name and the number of pages that form each rule. We also mention the class of visitors in which the rule was found. The rules listed in Table 5 are the navigation milestones for each class of visitors. Now, those rules can be used to simplify the representation of sessions. That process is explained next in Section 3.

**Table 5.** Name of the rules found in each class of visitors. The rule length indicates the number of pages that form each rule. An "X" indicates that the rule was found in that class of visitors.

| Rule Name | Rule Length | Made Payment | Started Payment | Other Conversion |
|---|---|---|---|---|
| Go to control panel | 4 | X | X | X |
| Pay via control panel | 7 | X | | |
| Pay and modify product | 5 | X | | |
| Consult availability and pay | 8 | X | | |
| Modify product and pay | 8 | X | | |
| Start payment | 4 | X | | |
| Pay for a service | 8 | X | | |
| Make invoice | 4 | X | | |
| Login and modify product information | 4 | X | | X |
| Modify product information | 3 | | X | X |
| Consult payment details | 3 | | X | |
| Make payments query | 3 | | X | |
| Modify product information and start payment | 3 | | X | |
| Consult availability | 3 | | | X |

## 3. Representation of Sessions with Rules

At this point, we have already identified the rules for each class of visitors. These rules can be used for representing sessions. This reduced representation allows us to carry out different kinds of analyses. In Section 3.1, we explain how we select rules to create a reduced graph and we provide statistics of the sessions represented with rules. These statistics provide information that would help marketing or information technology experts to establish questions of interest.Then, in Section 3.2, we describe how we select the data to be shown based on the questions of interest. For our particular case study, it was relevant to represent the navigation behavior of visitors only with the most frequent rules.

### 3.1. Selection of Rules to Visualize

We used the rules identified in Section 2 to represent sessions. In each session or group of sessions, we only used the rules that belong to the same class of visitors, according to Table 5. For example, a session that belongs to the class of visitors "Made payment" is represented only with the nine rules found in the sessions of the same class of visitors. In Figure 4, we show an example of three representations of a session: (1) the original session, (2) the session we obtained by replacing in the original session frequently occurring sub-sequences with rules (we called it a shrinked session), and (3) the session we obtained by stripping off any symbol but a rule in a shrinked session (we called it a stripped session).

The rules represent the behavior that is common among visitors from the same class. Conversely, pages that do not form rules represent uncommon behavior. To determine which analysis is worth conducting, we obtained statistics about sessions represented with rules. These statistics would help a marketing or information technology expert to determine questions of interest. We computed statistics on the three representations exemplified in Figure 4: original session, shrinked session, and stripped session. We computed the length of each representation and the reduction rate with respect to the length of the original session. Using the example in Figure 4, the length of the original session is 12, the length of the shrinked session is 5, and the length of the stripped session is 3.

- The reduction rate of the shrinked session is equal to $1 - ($length of the shrinked session/length of the original session$)$; that is, $1 - (5/12) = 0.58$. The length of the session is reduced by 0.58 (58%) when it is expressed with rules and pages that do not form a rule.

- The reduction rate of the stripped session is equal to $1 - ($length of the stripped session/length of the original session$)$; that is, $1 - (3/12) = 0.75$. The length of the session is reduced by 0.75 (75%) when it is expressed only with rules.

For each class of visitors, we computed the length of the three representations and the reduction rate. As an example, in Table 6, we show the results for the class of visitors "Made payment". We can see that the average reduction rate is 0.54 in shrinked sessions. For stripped sessions, the average reduction rate is 0.95. To better understand how the reduction rate behaves, we obtained a histogram of the reduction rate. In Figure 5, we show the histogram for shrinked and stripped sessions.

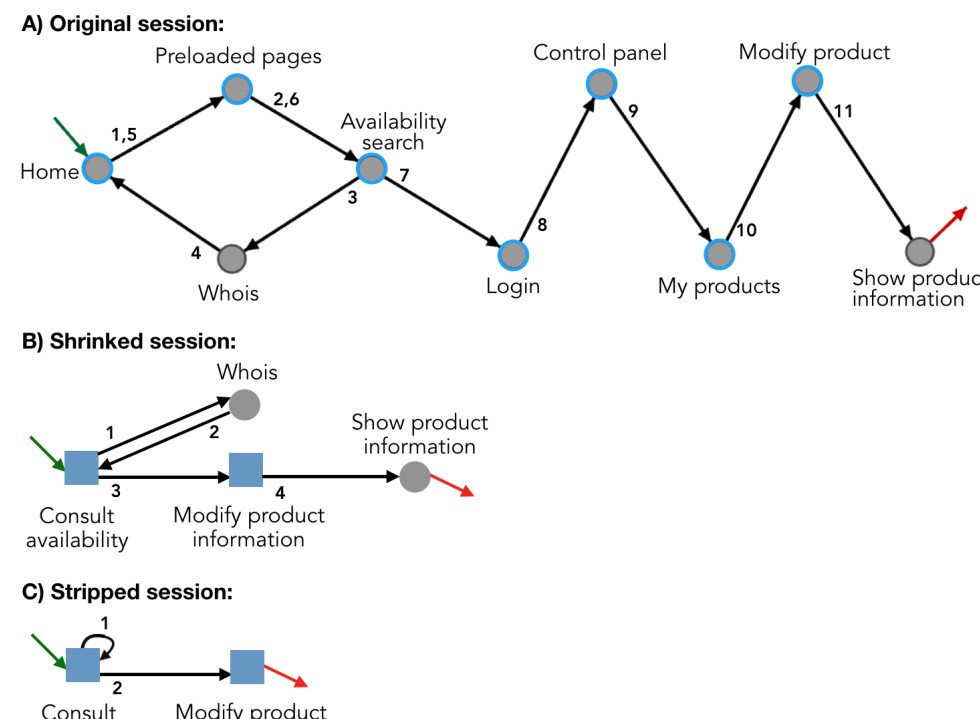

**Figure 4.** Example of a session represented with rules. (**A**) shows the original session. The green (respectively, red) arrow indicates the entry (respectively, exit) page. Circles with a blue border are pages that are part of a rule. This session belongs to visitors from the class "Other conversions". Therefore, we only used the rules identified in that class of visitors. We obtained (**B**) by representing the session with rules. (**C**) is the result of removing all pages that do not form a rule.

**Table 6.** Statistics of the length of sessions represented as rules. Metrics in rows 1 to 3 refer to the length of sessions in each representation. Metrics in rows 4 and 5 refer to the reduction rate with respect to the original session. In the last row, we indicate the percentage of sessions that we used in calculations. A total of 30% of sessions do not include any rule. Thus, in the last column, the percentage is reduced to 70%.

| Metrics for Visitors from the Class "Made Payment" | Original Session | Shrinked Session | Stripped Session |
|---|---|---|---|
| Average session length | 39.23 | 18.26 | 1.67 |
| Maximum session length | 572 | 181 | 24 |
| Standard deviation of session length | 30.81 | 16.46 | 1 |
| Average reduction rate from original session | NA | 0.54 | 0.95 |
| Standard deviation of reduction rate from original session | NA | 0.17 | 0.04 |
| Percentage of sessions considered | 100% | 100% | 70% |

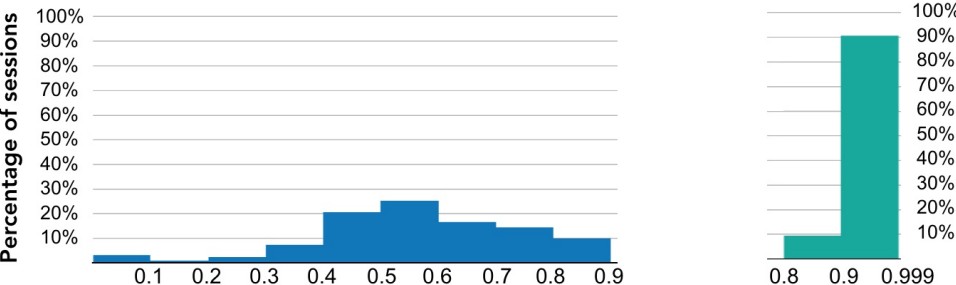

**Figure 5.** Histogram of the reduction rate of visitors from the class "Made payment". The reduction rate is calculated with respect to the length of the original session. The blue histogram corresponds to the shrinked sessions; their reduction rate varies from 0 to almost 1. The green histogram corresponds to the stripped sessions; their reduction rate varies from 0.8 to almost 1.

### 3.2. Selection of the Session Representation to Visualize

Previous statistics would help marketing or information technology experts to determine questions of interest; for example: what is the common navigation behavior in each class of visitors? what is different in the navigation behavior of each class of visitors? what navigation behavior is common among all classes of visitors? what are the relevant entry and exit milestones (rules) in each class of visitors?, etc. Different session representation is useful for each question.

In our case, the objective is to capture the milestones of different classes of visitors in a reduced representation of their navigation behavior. Therefore, for further analysis, we used the shrinked sessions. These allowed us to reduce the amount of data to analyze and contrast different classes of visitors. The elimination of pages that do not form rules does not affect our objective. On the contrary, including them would introduce information about the individual navigation behavior of visitors. Nevertheless, the analysis of pages that do not form rules could be relevant for other purposes; for example, to find out what distinguishes visitors from the same class.

Using shrinked sessions, we created a graph visualization that allows us to summarize the navigation behavior of each class of visitors. That visualization is presented next in Section 4.

## 4. Results

Shrinked sessions contain the milestones of the navigation behavior of visitors. In this section, we present those shrinked sessions in a graph visualization aimed at our target audience, marketing experts. Our work was not focused on visualization techniques, but the use of a graph is user friendly. It also assists in analyzing and comparing different rules or different classes of visitors. In Section 4.1, we explain how we built the graph. In Section 4.2, we describe the visualization of a whole class of visitors. Then, in Section 4.3, we exemplify the analysis of a single rule. Finally, in Section 4.4, we contrast different classes of visitors.

### 4.1. Graph Creation

In this subsection, we describe the concepts and calculations that we used to build a graph that describes the navigation behavior of visitors.

#### 4.1.1. Definitions

Consider $i$ and $j$ rules in the class of visitors $A$:

- Entry rate of the rule $i$: this is the number of sessions that start in the rule $i$ divided by the total number of sessions in the class $A$. It is denoted by $r(e_i)$.
- Exit rate of the rule $i$: this is the number of sessions that end in the rule $i$ divided by the total number of sessions in the class $A$. It is denoted by $r(x_i)$.

- Frequency of the edge $e\{i, j\}$: this is the flow of visits from the rule $i$ to the rule $j$. It is equal to the number of occurrences of the edge $e\{i, j\}$. It is denoted by $f_{ij}$.
- Out-degree frequency of the rule $i$: this is the flow of visits that goes out from the rule $i$. It is the sum of edge frequencies in which the source rule is $i$ plus $r(x_i)$. It is denoted by $O_i$.
- Weight of the edge $e\{i, j\}$: this is the frequency of the edge $e\{i, j\}$ divided by the out-degree frequency of the source node $i$; that is, $\frac{f_{ij}}{O_i}$. It is denoted by $w_{ij}$.

### 4.1.2. Calculation Example

Consider the rules $a$ and $b$ found in the class of visitors "A". This class has 10 sessions. In addition, consider the following information:

- Entry rule: seven sessions started in rule $a$ and three sessions started in rule $b$.
- Exit rule: two sessions ended in rule $a$ and eight sessions ended in rule $b$.
- Edge frequency: $f_{ab} = 5$, $f_{aa} = 3$, and $f_{ba} = 6$.

In order to construct the graph, it is necessary to calculate the entry rates, the exit rates, and the weights.

- Entry rate: $r(e_a) = 7/10 = 0.7$ and $r(e_b) = 3/10 = 0.3$.
- Exit rate: $r(x_a) = 2/10 = 0.2$ and $r(x_b) = 8/10 = 0.8$.
- The calculation of weights is shown in Table 7.

**Table 7.** Example of weight calculation. The out-degree frequency $O_i$ is the sum of frequencies $f_{ij}$ of edges with the same source rule. Thus, $O_a = 5 + 3 + 2 = 10$ and $O_b = 6 + 8 = 14$.

| Source Rule $i$ | Target Rule $j$ | $f_{ij}$ | $O_i$ | $w_{ij} = \frac{f_{ij}}{O_i}$ |
|---|---|---|---|---|
| a | b | 5 | 10 | 5/10 = 0.5 |
| a | a | 3 | 10 | 3/10 = 0.3 |
| a | Exit | 2 | 10 | 2/10 = 0.2 |
| b | a | 6 | 14 | 6/14 = 0.43 |
| b | Exit | 8 | 14 | 8/14 = 0.57 |

### 4.1.3. Graph Example

In Figure 6, we show the graph obtained in our example.

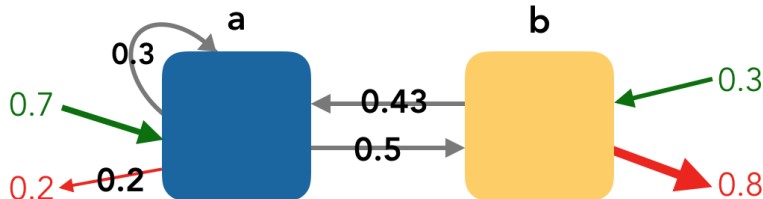

**Figure 6.** (**a**,**b**) Graph example. Yellow nodes are the rules where conversion occurs. The arrow thickness corresponds to the edge weight. The green (respectively, red) arrows indicate the entry (respectively, exit) rate. The values in the middle of the arrows indicate the weight of the edge ($w_{ij}$). If there were edges with a weight <0.05, they would be in a lighter grey, and their weight would not be shown.

### 4.2. Visualization of a Whole Class of Visitors

We created the graph for each class of visitors as described in Section 4.1. In Figure 7, we show the graph of visitors that belong to the class "Made payment". The marketing expert could determine if the observed behavior is expected or if there is suspicious or interesting behavior that is worth investigating. The interpretation of the graph depends on the business questions in which the marketing expert is interested. Below, we present our interpretation.

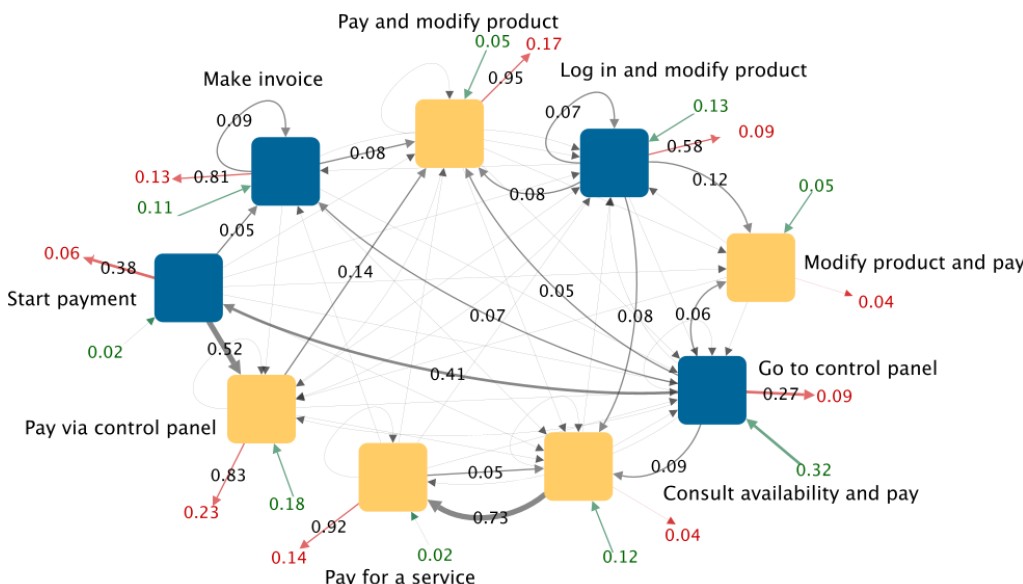

**Figure 7.** Graph of shrinked sessions for visitors from the class "Made payment". Yellow nodes are the rules where the payment is confirmed.

### 4.2.1. Relevant Entry and Exit Rules

In Figure 7, we can see that the rule "Go to control panel" has the highest entry rate (0.32). A total of 32% of the sessions have this rule as the entry point. The rule "Pay via control panel" has the highest exit rate (0.17). A total of 17% of the sessions have this rule as the exit point.

Based on the weight of the edges, there are three rules with a weight >0.90 on their red arrow. Those rules are "Pay and modify product", "Pay for a service", and "Modify product and pay". This means that almost all visitors who follow those rules leave the website after that. Contrarily, only 25% of visitors who follow the rule "Consult availability and pay" leave the website.

### 4.2.2. Most Frequent Path

If we follow the path of the highest entry rate and highest weights, we can see that 32% of visitors enter by the rule "Go to control panel". From there, 41% of visitors follow the rule "Start payment". Then, 52% of visitors follow the rule "Pay via control panel". In this rule, the payment is confirmed. After that, 83% of visitors leave the website. The marketing expert could evaluate if this path was expected. For example, is that sequence of 15 pages adequate? Could it be shorter? Was it expected that visitors leave the website immediately after the payment?

### 4.2.3. Rules in Which Conversion Occurs

From all of the rules in which the payment occurs, most visitors leave the website. The marketing expert could determine feasible strategies to retain visitors after a purchase; for example, a flash discount on the purchase of additional service. There is an exception in the rule "Consult availability and pay". From this rule, 73% of visitors continue with the rule "Pay for a service". Those visitors made two payments because, in both rules, the payment is confirmed.

Besides observing the "big picture" in the graph, it is also possible to analyze specific rules in more detail. Next, we exemplify it.

### 4.3. Analysis of Specific Rules

From the rules in which the payment does not occur, the rule "Make invoice" has the highest exit rate. Therefore, we will analyze this rule further. Visitors who follow this rule mainly come from the rules "Start payment" and "Go to control payment". In those

rules, the payment has not been confirmed. Additional analysis from marketing expert is needed to determine the reasons for the described behavior. For example, is the making of the invoice clear? Is it a long process? Does it have an annoying bug? Does it require redundant information? Is it used by clients or competing companies to inquire about the prices of products or services?

After knowing the reasons for losing visitors in the rule "Make invoice", marketing experts could design strategies for retaining them; for example, proactive online help. If the marketing team does not have enough information to determine why visitors are leaving after this rule, different actions could be implemented; for example, a pop-up window to rate the process to make the invoice. Even users who confirm the payment may provide useful information about this process.

Besides reviewing rules in detail, the graph representation also allows us to compare different classes of visitors. This is exemplified next.

### 4.4. Contrasting Different Classes of Visitors

The comparison of different classes of visitors depends on the behavior in which the marketing expert is interested. Below, we present how we contrasted the classes of visitors "Made payment" (shown in Figure 7) and "Started payment" (shown in Figure 8).

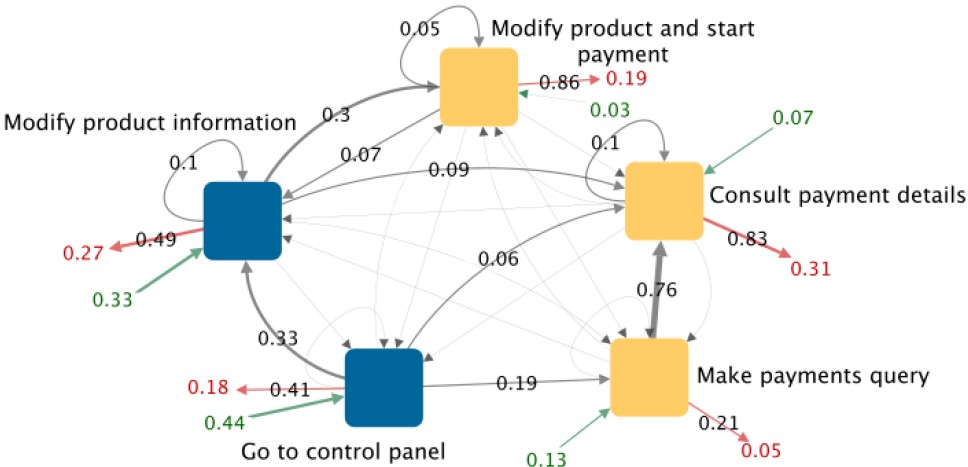

**Figure 8.** Graph of shrinked sessions for visitors from the class "Started payment". Yellow nodes are the rules where the payment is started.

### 4.4.1. Contrasting the Exit Rule

Most visitors from the class "Made payment" leave the website after following the rule "Pay via control panel". This is a rule in which the payment is confirmed. Most visitors from the class "Started payment" leave the website after following the rule "Consult payment details". This is a rule in which visitors start the payment. This indicates that most visitors who start a payment but do not confirm it leave the website immediately after consulting the payment details instead of navigating further or requesting online help.

In both classes, the highest exit rate is in the rules in which a conversion is performed, even though the conversion is different in each class of visitors.

### 4.4.2. Contrasting a Common Rule

In both classes of visitors, the rule "Go to control panel" has the highest entry rate. Nevertheless, the exit rate is double in visitors from the class "Started payment". After following the rule "Go to control panel", most visitors from the class "Made payment" start the payment process, whereas most visitors from the class "Started payment" modify the product information. This could be useful for encouraging the purchase in the pages where the product information is modified.

### 4.4.3. Contrasting the Most Frequent Path

The path with the highest entry rate and weights, in visitors from the class "Made payment", is "Go to control panel" (32%) → "Start payment" (41%) → "Pay via control panel" (52%) → Exit (83%). In visitors from the class "Started payment", the path with the highest entry rate and weights is "Go to control panel" (44%) → "Modify product information" (41%) → Exit (49%). This confirms the relevance of the rule "Modify product information" as an exit point.

### 5. Discussion

We presented the graph of shrinked sessions for different classes of visitors. This visualization reduces the amount of data that marketing experts would have to analyze for understanding the navigation behavior of visitors. It also assists in contrasting different classes of visitors. Our main contributions were: (1) to obtain results that are easy to interpret and could be meaningful for marketing experts. We achieved this by using classes of visitors associated with conversions of the sales funnel; and (2) to ease the analysis of results with a simplified description of the navigation behavior of visitors. This was created by using rules, which are the sequences of pages of common occurrence.

In Section 1.1 "Previous work", we describe previous commercial and non-commercial approaches related to our work. In Table 8, we summarize their differences with our method.

**Table 8.** Differences between our method and previous approaches.

| Type | Author | Differences with Our Approach |
|---|---|---|
| Commercial software | Google Analytics (GA) [11] and Matomo [12] | They provide graphs with page-by-page detail. This helps to measure web page engagement; for example, to find pages where traffic is lost. However, this detail makes it difficult to visualize a trajectory with numerous pages or to visualize 100 percent of visitors. It is possible to track events instead of web pages, but since events are pre-configured, they do not necessarily represent the natural navigation behavior of visitors. In addition, GA does not allow you to analyze data generated prior to its use. |
| Non Commercial software | S. Tiwari et al. [20] | Its goal is to find the expectation of the next page using agglomerative clustering. Visitors are classified based on previous web pages they accessed, but web pages are not clustered or associated with conversions. Therefore, classes are not necessarily meaningful to a business expert and do not provide a high-level representation of the navigation behavior. It is useful in an online implementation to improve web response time, but is not intended to facilitate analysis for MKT experts. |
| | A. Banerjee et al. [27] | Its purpose is to find the longest common sub-sequence of clickstreams using a dynamic programming algorithm. They use a similarity graph to find clusters of visitors based on the time spent on each page. In our approach, the longest path is not necessarily the most relevant in terms of business goals. In fact, we found that, in general, visitors who make a payment visit fewer pages than those who do not. In our approach, the time spent on each webpage is not relevant. Finally, the similarity graph that they build is aimed to cluster visitors but is not to be used by marketing experts for further analysis. |
| | Huy M. et al. [21,22] | They present a novel data structure (pseudo-IDList) suitable for clickstream pattern mining. They also propose using the average weight measure for clickstream pattern mining and present an improved method named Compact-SPADE. Both approaches focus on improving runtime or memory consumption for clustering visitors, but no business knowledge is incorporated to create those clusters. In addition, they do not create a high-level graph of the clusters for further analysis by business experts. |
| | F. Gómez [34] | He presents a visualization tool for analyzing website traffic. It aims to distinguish bots from human visitors based on their navigation path. Like commercial software, it provides a page-by-page detail, which makes it difficult to analyze numerous pages. |
| | B. Cervantes [35] | They combine visualization and machine learning techniques for analyzing web log data. The visualization can be used by business experts to obtain insights by looking at key elements of the graph. However, visitors are not classified in terms of business goals. Furthermore, the visualization provides page-by-page detail, which makes it difficult to analyze visits with numerous nodes. |

Web analytics software is essential for measuring website traffic and follow-up marketing campaigns. Nevertheless, its standard configuration and reporting options make it hard to extract high-level knowledge about the navigation behavior of different classes of visitors, especially due to the high amount of website traffic and the diversity of paths that visitors could follow. Most non-commercial approaches focus on finding clusters of visitors or improving the runtime performance, and proposed visualizations also provide a page-by-page detail that may lead to enormous graphs that are difficult to analyze. To highlight the advantages of the method that we propose, we compared the resultant visualization (Figure 8) with the visualization of Google Analytics (Figure 2) and a non-commercial approach (Figure 9). We will refer to the two last as CSW (commercial software) and NCA (non-commercial approach), respectively. Next, we list the most relevant differences:

- Both CSW and NCA provide page-by-page detail. Considering that most commercial websites have hundreds of pages, a graph that shows all website sessions in a given period would be (1) extremely long in CSW and (2) uninterpretable in NCA.
- Both CSW and NCA allow us to filter segments of visitors, but this is not enough to have an easy-to-interpret graph. CSW allows us to select the starting webpage in the graph. However, that leads to an incomplete graph. It allows for an analysis of engagement in the selected web pages but does not allow for a visualization of the complete path followed by visitors.
- CSW allows tracking pre-configured events instead of web pages but without the context of all of the web pages visited by the related visitors. Conversely, our approach makes no assumptions about the visitor behavior and provides the rules (sequences of visited pages) in the context of the complete navigation paths.
- Neither CSW nor NCA eases the identification of loops. Our proposed method clearly shows loops, in a single conversion and between different conversions.
- In CSW and NCA, it is hard to visualize the navigation behavior of a whole class of visitors. Therefore, it would be more difficult to compare different classes of visitors. Conversely, our approach would provide a moderately small graph for each class of visitors, and these graphs are easier to compare. This comparison of classes allows us to answer specific business questions. For example, what is the most common sequence of visited pages on which visitors of two different classes leave the website?
- Generally, achieving business objectives (conversions) involves the visiting of several pages. Some business goals may be partially identified as events in CSW, e.g., the sequence of web pages that a visitor must follow to make a payment. However, visitors may follow longer common sequences that help to reduce the number of nodes. In addition, some common navigation behavior is not predictable. For example, what do visitors who did not finish the payment process have in common?

Our method helps to identify points of interest whose interpretation is enriched by the opinion of a business expert. This approach assists in answering business questions in the context of the navigation behavior of all visitors in a given class, which is opposite to existing solutions that mainly aim to analyze the web page performance in detail. Next, we mention some examples of findings that would be difficult to obtain in a graph with page-by-page detail:

- The identification of unexpected loops or repetitions that could be avoided with website enhancements. For example, in Figure 7, we can see that 5% of the traffic in the node "Modify product and start payment" loops in this rule. The business expert could investigate the cause, e.g., a technical error, a non-intuitive site design, or a lack of clarity in the information shown.
- Identification of processes in which web traffic is lost. For example, in Figure 7, we can see that there are three rules in which the payment process can start: (1) "Modify product and start payment", (2) "Consult payment details", and (3) "Make payments query". However, the dropout rate is four times higher in the first two and they

have a loop. A call to action in the web pages of rules 1 and 2 could decrease their dropout rate.

- Finding the relationship between two conversions, regardless of whether they are subsequent or not. For example, we found that a high percentage of visitors who dropped out before confirming the payment did not request assistance at any point in their session. Thus, the online helpdesk is underutilized, as it could help to retain customers that leave before payment is finalized. A call to action on the pages prior to payment confirmation could increase the conversion rate.
- Comparing different website versions or different periods at a high level. If a website changes drastically, it can be difficult or even impossible to compare the impact of each page on the user experience. With our method, we can compare the versions at a high level. We could find, for example, how many steps the most common path has and if the new version reduced or increased the loops in navigation, and could compare the shortest path to make a payment versus the most common path to pay.
- Contrasting the navigation behavior of different classes of visitors. For example, new vs. recurrent; male vs. female; or visitors from different countries. All of them are in the context of the whole navigation behavior of the classes of interest. For example, the graphs in Figures 7 and 8 can be compared as discussed in Section 4.4. This comparison would not be easy in two graphs with hundreds of nodes and edges.

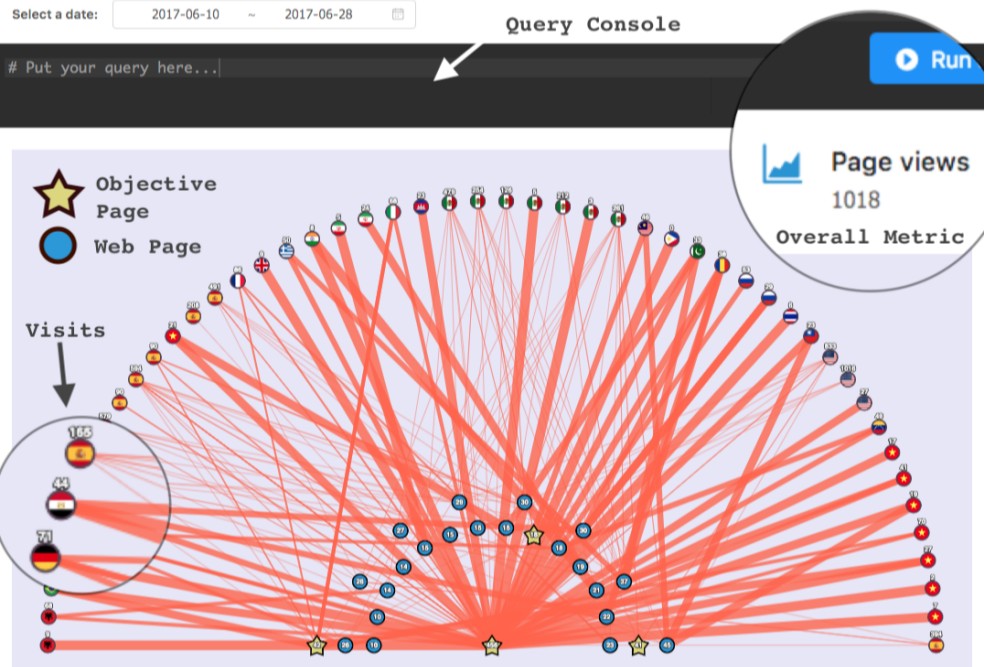

**Figure 9.** An example of the visualization proposed by B. Cervantes et al. [35]. Blue nodes are web pages, starts are objective pages, and nodes with country flags are visits of users from that country.

## 6. Conclusions

To describe the navigation behavior of visitors, we proposed a clickstream analysis. It is based on identifying actions that are repeated by users of the same class, considering an action as a sequence of visited pages. To assist in the interpretation of results to marketing experts, we created a graph representation. Our proposal is a starting point to further simplify the analysis of the navigation behavior of visitors or the extraction of knowledge for a marketing audience. Next, we summarize the contributions of our method, its limitations, and future work.

### 6.1. Contributions

There are three main advantages of our methodology over other existing solutions. The first advantage is that it reduces the amount of data to analyze for understanding the navigation behavior of visitors. The second advantage is that it extracts the natural navigation behavior of visitors. The third advantage is the use of web logs as entry data. Below, we explain them in more detail.

The increasing amount of data generated on websites makes it difficult to find relevant knowledge. With our method, we replace tens of pages with a single graph. Besides summarizing the navigation behavior of a whole class of visitors, our approach also allows us to compare different classes of visitors. This knowledge could be used to improve the effectiveness of marketing campaigns or website design. For example, an action (which is composed of a sequence of pages) could be especially successful to attract new visitors, but unsuccessful to make clients purchase. Marketing experts could design strategies for visitors to leap from the interest stage to the purchase stage (e.g., add proactive online help, provide more information about the benefits of the product, or a retargeting campaign for the visitors who performed visits of that sequence of pages).

With our methodology, we extracted the natural navigation behavior of visitors. This distinguishes our work from other solutions. Previous approaches group the web pages in tasks identified by the business expert. Therefore, they reflect the expected behavior, not the natural paths followed by visitors. Our approach, on the contrary, obtains the common sub-sequences of visited pages that visitors follow. This approach allowed us to find useful information; for example, that the making of the invoice is a relevant exit point. It also enabled us to contrast relevant entry and exit rules for different classes of visitors.

The use of web logs as entry data allows for the performance of a retrospective analysis. This is not possible in commercial software, where the configuration of conversions and market segments usually applies only for future traffic of the website. The use of web logs also allows us to prepare data according to different objectives. For example, we could compare different periods of a given class of visitors or different website versions.

### 6.2. Limitations

Our method focuses on a high-level understanding of navigation behavior. However, some business questions necessarily require a detailed web-page-level review. For those cases, commercial software is already effective; for example, if we want to know where traffic flows to after the visit of a specific web page.

Our approach can extract facts; for example: out of N possible ways to make a payment, which one is the shortest or the most frequent. However, in other cases, the findings are only the beginning of the discussion and require the intervention of a business expert. For example, a loop is not necessarily bad, but the interpretation of a human expert is needed to identify which ones are worth analyzing and correcting.

The effectiveness of our method relies on the existence of sequences of pages that are common among visitors to a website. However, there is a possibility that the navigation is too sparse for a particular class of visitors. In those cases, there will be no rules with a relevant frequency. While this would in itself be a finding, it would not be possible to construct a graph for further analysis by a business expert.

### 6.3. Future Work

There is a latent need for creative ways to help business experts evaluate the performance of a website. Next, we mention a few examples of how our method could be improved or extended:

- The effectiveness of our approach could be tested for improving metrics measurement, website design, and paid marketing effectiveness.
- Software aimed at marketing experts could be useful for autonomously replicating and personalizing the process that we followed. For example, the company could identify the five most relevant conversions and use them for describing the navigation

behavior of visitors. On the contrary, the company may find it useful to associate each page of interest with a conversion.

- It would be valuable to extract rules from the high-level visualizations that we obtained. For example, Acosta-Mendoza et al. propose a frequent approximate subgraph mining approach [51], which we could incorporate as the last step of our methodology.
- The use of rules that we propose can be applied after classifying visitors with different methods. We used conversions to classify visitors because we focused on a marketing audience. However, visitors could be classified with other techniques and purposes.
- After identifying rules of interest with our method, some of them could be configured in web analytics software for monitoring (e.g., such as events in Google Analytics). Although visitor behavior is dynamic, it could be useful for the marketing expert to monitor some rules autonomously.

Our approach responds to the need for a high-level description of the navigation behavior of website visitors. It does not replace the functionality of existing web analytics software; on the contrary, it can complement it. Our method can also be used with existing classification techniques. This work is a starting point for business questions that require understanding the navigating behavior of website visitors in a wide context. We believe that this is a very promising area of research.

**Author Contributions:** Conceptualization, R.M. and B.C.; methodology, A.H.; software, A.H.; validation, A.H.; formal analysis, A.H.; investigation, A.H.; resources, R.M. and B.C.; data curation, A.H.; writing—original draft preparation, A.H.; writing—review and editing, R.M. and B.C. visualization, A.H.; supervision, R.M. and B.C.; project administration, A.H. and R.M.; funding acquisition, R.M. All authors have read and agreed to the published version of the manuscript.

**Funding:** The research reported here was supported by Consejo Nacional de Ciencia y Tecnología (CONACYT) studentship 957562 to the first author.

**Institutional Review Board Statement:** Not applicable.

**Data Availability Statement:** We did not use publicly available datasets. We thank NIC Mexic for providing the data used in this research.

**Acknowledgments:** The authors acknowledge the technical support of Tecnologico de Monterrey, Mexico. We also thank NIC Mexico for providing the data used in this research.

**Conflicts of Interest:** The authors declare no conflict of interest.

## Abbreviations

The following abbreviations are used in this manuscript:

| | |
|---|---|
| WAS | Web analytics solutions |
| CSW | Commercial software |
| NCA | Non-commercial approach |

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
