# Peer review of "A High-Level Representation of the Navigation Behavior of Website Visitors"

_applsci, doi:10.3390/app12136711_

Round 1

Reviewer 1 Report

My specific comments would be as below:  

This is a very interesting and well-written manuscript that I really enjoyed reading! The caption of Table 2: Columns 2 to 4 indicate the class of visitor. This should be columns 2 to 5!  

It is also possible to combine figures 4 and 5 into one figure with two sub-figures. This makes the comparison more easily.  

The authors of the manuscript mentioned a few times that standard configuration and reporting options of commercial software may make it difficult to extract high-level information about the navigation behavior of different groups of visitors. I would recommend showing at least one example of how the navigation behavior of visitors differs between the proposed method and something else, for example, Google Analytics.  

In Section "Discussion", It would be helpful to discuss the obtained results in more detail. The majority of the results discussed in the paper could be estimated before doing the analysis. Which observation was interesting? How can the graph representations be useful in analyzing the behavior of customers?

Reviewer 2 Report

Brief summary:
The paper presents a description of the website visitors’ navigation behavior based on a clickstream analysis. The results are mainly useful for marketing experts.

General comments:
The paper contributes to practical application in the area of marketing and website analytics. However, there is a lack of scientific context in Discussion and Conclusions.

Specific comments:
Discussion (L496-508):
It is recommended to compare the authors’ results with other similar studies in this scientific area. The authors describe their own contributions, but this description would be more credible if their results were presented in relation with other researchers’ work.

Conclusions (L551-553):
At the end of Conclusions the authors mention the suggestion for future work. Even so, it would be also appropriate to specify the limitations of the methodology presented in this paper.

Other minor changes:
L134-135: If the authors think that this sentence is necessary, than it should contain the section number.

The reference list should be written more carefully. Some references are incompletely cited.
